



# Short communication: Concentrated impacts by tree canopy drips: hotspots of soil erosion in forests

Ayumi Katayama[1], Kazuki Nanko[2], Seonghun Jeong[3], Tomonori Kume[1], Yoshinori Shinohara[4], Steffen Seitz[5]

[1]Shiiba research forest, Kyushu University, Miyazaki, 8830402, Japan
[2]Department of Disaster Prevention, Forestry and Forest Products Research Institute, Tsukuba, 3058687, Japam
[3]Department of Forest Environmental Resources, Gyeongsang National University, (Institute of Agriculture & Life Science), Jinju, 52828, Republic of Korea
[4]Faculty of Agriculture, University of Miyazaki, Miyazaki, 8892155, Japan
[5]Department of Geosciences, University of Tübingen, Tübingen, 72074, Germany

*Correspondence to*: Ayumi Katayama (ayumi.katayama0920@gmail.com)

**Abstract.**

The degradation of ground vegetation cover caused by large grazing herbivores frequently results in enhanced erosion rates in forest ecosystems. Splash erosion can be caused by drop impacts with high throughfall kinetic energy (TKE) from the canopy of the trees. Notably bigger canopy drips from structurally-mediated woody surface points appear to induce even higher TKE and generate concentrated impact locations causing severe focus points of soil erosion. However, TKE at these locations has rarely been reported. This study investigated the intensity of TKE at a concentrated impact location and compared it to general TKE locations under the canopy and freefall kinetic energy (FKE) outside the forest. We measured precipitation, TKE and FKE using splash cups at seven locations under Japanese beech trees and five locations outside the forest in the leafless and leafed seasons in a deciduous broadleaved forest of Japan, respectively. TKE at the concentrated impact location was 15.2 and 49.7 times higher than that at general locations under beech and FKE, respectively. This study confirmed that canopy drip from woody surfaces can be a hotspot of soil erosion in temperate forest ecosystems. Throughfall precipitation at the concentrated impact location was 11.4 and 8.1 times higher than that at general locations and freefall, respectively. TKE per 1 mm precipitation (unit TKE) at the concentrated impact location ($39.2 \pm 23.7$ J m$^{-2}$ mm$^{-1}$) was much higher than that at general locations ($22.0 \pm 12.7$ J m$^{-2}$ mm$^{-1}$) and unit FKE ($4.5 \pm 3.5$ J m$^{-2}$ mm$^{-1}$). Unit TKE in the leafless season was significantly lower than in the leafed season because of fewer redistribution of canopy drips induced only by woody tissue. Nevertheless, unit TKE at the concentrated impact location in the leafless season ($36.4$ J m$^{-2}$ mm$^{-1}$) was still higher than at general locations in the leafed season. These results show that potentially high rates of sediment detachment can be induced by not only throughfall precipitation, but also larger throughfall drop size distributions at concentrated impact locations, even in the leafless season.



## 1. Introduction

Soil conservation is an important environmental challenge of the 21st century as soils are the foundation of life and a reservoir for water, carbon, and nutrients (Lal, 2014). Worldwide, they are still endangered in their substance, especially in areas with regularly recurring climatic extreme events such as heavy rainfalls (Borrelli et al., 2020). Soil erosion rates induced by water are mainly determined by rainfall patterns such as raindrop kinetic energy and ground cover by vegetation (Seitz et al, 2017). In forest ecosystems, severe soil erosion events are rare as abundant ground cover is generally occurring through understory vegetation or plant litter (Miura et al. 2003; Holz et al, 2015). Therefore, forest can be seen as one of the most effective land use types to mitigate soil losses (Pimentel and Burgess, 2013). However, disturbance of forest vegetation may lead to significant punctual (Gall et al, 2022; Geißler et al, 2010) and areal (Safari et al, 2016; Seitz et al, 2016; Zemke et al, 2016) erosion events that can by far exceed sustainable erosion rates (Deng et al. 2023). Important examples have been described globally such as in Hungary (Misik and Kárász, 2022) and China (Yao et al., 2019). Especially in Japan, understory vegetation in forests is regularly damaged by grading sika deer (*Cervus nippon*) (Murata et al., 2009, Takatsuki 2009). The degradation of protective vegetation layers frequently results in enhanced splash erosion through direct raindrop impact and increased surface runoff with significant erosion potential (Shinohara et al, 2018; Song et al, 2019).

Throughfall kinetic energy (TKE, in J m$^{-2}$) is determined by drop size and velocity in addition to precipitation amount. TKE has partly shown to be higher than freefall kinetic energy (FKE) outside vegetation layers as forest canopy can generate large new canopy drips after the first interception depending on the species (Chapman, 1948; Nanko et al., 2015). Canopy drip can contribute to more than half of total throughfall in volume from leafed canopies (Levia et al, 2019). In canopy water flow, the lateral redistribution plays an important role in creating local concentration of throughfall (Keim and Link, 2018). Subsequently, lateral canopy water flow paths ending at structurally-mediated woody surface drip points, such as irregular rough points and branch concavities, accumulates more water volume transported down the branch with a longer residence time and then generate larger diameter drops in greater volumes (Nanko et al.,2022) than foliar surfaces (Levia et al., 2019; Nanko et al., 2016; Nanko et al., 2022). Notably bigger canopy drips can have higher TKE and therefore, generate concentrated impact locations potentially causing severe soil erosion. However, the TKE at these concentrated impact locations and subsequent splash erosion potential has only rarely been described in literature and not been quantified yet.

TKE is linearly correlated with throughfall precipitation in monolayer coniferous forests (Shinohara et al., 2018). The slope of the relationship between throughfall precipitation and TKE is known as unit TKE, that is, TKE per 1 mm precipitation. Previous studies showed that the unit TKE differed with canopy species and architecture, and rainfall intensity (Nanko 2013, Nanko et al., 2015, Liu et al., 2022). Throughfall from woody surface drip points consist of larger canopy drips, suggesting the unit TKE at such concentrated impact locations being different from that at other general locations.





Furthermore, this relationship might also differ between leafed and leafless seasons where drop size
distributions have proven to be varying (Levia et al., 2017). Thus, TKE can considerably affect soil
erosion rates also in the leafless season when the contribution of drip points to total throughfall
precipitation becomes dominant (Levia et al., 2019). Therefore, knowledge about significance of TKE at
concentrated impact locations and seasonal changes in TKE in response to leaf status is vital for
understanding soil erosion risk in forests with degraded ground cover.

80         This study reports TKE under broadleaved trees in Shiiba research forest, Kyushu, Japan, a

strongly disturbed and eroded forest ecosystem due to deer grazing. A special focus of this study is given
on unusual high energy levels induced by structurally-mediated woody surface drip points which partly
occurred during the measurement campaign with splash cups to estimate throughfall erosivity. In this
study, the intensity of TKE at this concentrated impact location was quantified.

85         It is hypothesized that (1) unit TKE at the concentrated impact location is higher than that at

general locations inducing elevated splash erosion, and (2) the relationship between throughfall
precipitation and TKE differs with the leaf status of trees.

**2. Materials and methods**
**2.1 Study site**
This study was conducted in Shiiba research forest, Kyushu, South Japan [32°40′N, 131°17′E, 1030 m
a.s.l.]. Here, mixed forest with evergreen coniferous trees and deciduous broadleaved trees can be found.
The mean annual temperature and precipitation are 10.8°C and 3278 mm, respectively, which were
measured at a meteorological station located 3 km from the study site at 1180 m a.s.l. The area was
formerly characterized by dense bamboo (*Sasa borealis* [Hack.] Makino & Shibata) vegetation at the
understory. However, this understory vegetation has mostly disappeared since around the year 2000, as an
increase in Sika deer population was registered. Today, there is no intact understory vegetation in most of
the area of the research forest (Kawakami et al, 2020). Therefore, distinct erosion forms and root
exposure can be observed widely and soil degradation has been pointed out a major challenge for the
forest service (Abe et al. 2022).

**2.2 Throughfall kinetic energy**
TKE was determined as a proxy for splash erosion using splash cups (Shinohara et al., 2018; Scholten et
al., 2011). Splash cups are filled with a standardized sand and weighed in dry before deployment in the
field. Raindrops subsequently hit the sand surface and detached sand is partly splashed away from the
cup. The loss of sand (LoS, g m$^{-2}$) is measured by back weighing remaining dried sand volumes and
subtracting the amount from the initial amount. TKE can be estimated from the relationship between KE
and LoS using a linear function (TKE = 14.55 × LoS, Scholten et al., 2011). This method has proven to
be reliable and cost efficient with a high number of replications (Geißler et al., 2010) and is suitable to
evaluate spatial variation in TKE (Shinohara et al., 2018).



LoS was measured during each five rainfall events in the leafless (March to April) and the leafed
(August to September) season in 2021. Seven splash cups were installed under the canopy of two *Fagus*
*crenata* trees for TKE. One position was chosen at a possible concentrated drip location formed by
structurally-mediated wood surface, and where more throughfall precipitation was observed by eye during
rainfall events. Six more splash cup positions under the canopy were installed to measure TKE at general
locations. Five splash cup positions were further selected outside the forest to measured FKE. A rainfall
collector was installed next to each splash cup to quantify precipitation at the measuring location.
At the concentrated impact location, the collection of LoS and throughfall precipitation missed
for some very strong rainfall events during the leafed period. Deployed splash cups were either emptied
completely (three events) or the throughfall collectors overflowed (four events), indicating the
extraordinarily high TKE. For these rainfall events, TKE and throughfall precipitation were estimated
from the relationship between TKE and freefall precipitation (TKE = 237.1 × freefall precipitation, $R^2$ =
0.92) and throughfall and freefall precipitation (throughfall precipitation = 8.23 × freefall precipitation, $R^2$
= 0.97) obtained in other events.

**2.3 Tree traits**
Diameter at breast height of the two selected beech trees were 46.0 cm and 46.1 cm, and tree height
was 21.1 m and 18.0 m, respectively. LAI determined with a single reflex camera system with fish eye
lens (THETA SC; Ricoh Co. Ltd., Tokyo, Japan) and software (a Gap Light Analyzer ver. 2.0, Frazer et
al., 2022) was 4.5 and 0.9 at the concentrated impact location in the leafed and leafless season,
respectively. LAI at general locations ranged from 1.7 to 4.9 with a mean of 3.3 and from 0.1 to 0.6 with
a mean of 0.3 in the leafed and leafless season, respectively. Branch height at the concentrated impact
location was 9.1 m and ranged from 6.5 m to 13.5 m with an average of 9.1 m at the six splash cup
positions. Average leaf area and leaf mass per area obtained from beech leaves in our study forest were
10.5 cm$^2$ and 84.7 g m$^{-2}$, respectively. The bark of the beech was smooth, but there was moss cover in
some places along the stem and epiphytic moss at the base of the branch, from which considerable
amounts of water dropped to the ground.

**2.4 Statistical analysis**
The significant difference in slopes in the relationships of throughfall precipitation with TKE between
concentrated impact location and general locations was examined using ANCOVA ($P < 0.05$). The
significant difference in slopes in the relationships between leafed and leafless seasons was examined for
impact and general locations separately (ANCOVA, $P < 0.05$). In these analyses, TKE data which was
not measured in the three rainfall events was excluded. The intercepts were set at zero in the models. All
statistics were performed in *R* ver. 3.6.2 (*R* Core Team, 2019).

**3. Results and Discussion**
**3.1 Effect of structurally designed high energy points on TKE**



Considerable high TKE was observed at the concentrated impact location under the beech (Fig. 1). The
location received a focused number of canopy drips from an overlying structurally-mediated woody
surface drop point (supplemental video). Average ± S.D. of TKE at the concentrated impact location
$(9142 \pm 5522$ J m$^{-2})$ for all seasons was 15.2 times higher than at general locations under the beech $(601 \pm$
495 J m$^{-2})$ and 49.7 times higher than FKE (184 ± 195 J m$^{-2}$, Table 1) underlining the important TKE-
increasing potential of tree traits such as branch height and leaf size (e.g., Geißler et al, 2012; Goebes et
al, 2015). The average of throughfall precipitation at the concentrated impact location (324 ± 227 mm)
was 11.4 times higher than that at general locations under beech (29 ± 16 mm) and 8.1 times higher than
that from freefall precipitation (40 ± 26 mm).

158        Across all rainfall events, TKE significantly increased with throughfall precipitation at both the

concentrated impact location and general locations regardless of canopy leaf conditions (Fig. 2). The  It
could be shown that TKE at the concentrated impact location was strongly higher than at general
locations with a significant difference in the relationships between TKE and throughfall precipitation
(Fig. 2). Thus, the first hypothesis can be confirmed. Furthermore, the branch height at the concentrated
impact location was comparable to average of branch height at other general drip points, indicating that
higher unit TKE was mostly induced by bigger drop sizes. Note that the unit TKE is determined from
raindrop size distributions and canopy height when the canopy height is less than the height for the rain-
drop terminal velocity (Shinohara et al., 2018). Previous study showed that most canopy drips did not
reach to the terminal velocity where the mean first living branch height was 7.9 m (Nanko et al., 2008).
Raindrops with diameters >3 mm need at least 12 m fall distance to gain terminal velocity (Wang and
Pruppacher, 1977). Thus, the TKE at the concentrated impact locations originating from woody surface
was induced by both high throughfall precipitation and big drop size, which is an important cause of
splash erosion and might be considered as an underestimated hot spot of sediment translocation.

**3.2 Effects of leaf status**
In the leafed season, event-scale average TKE at the concentrated impact location was 12.5 times higher
than those at general locations under the beech tree and 61.5 times higher than FKE (Table1). Event-scale
mean throughfall precipitation at the concentrated impact location was 12.2 times higher than at general
locations and 8.1 times higher than freefall precipitation. In the leafless season, the average TKE at the
concentrated impact location was 23.6 times higher than those at general locations and 37.6 times higher
than FKE, whereas mean throughfall was 10.3 times higher at general locations and 8.2 times higher than
freefall precipitation. These results suggest that splash erosion risk at the impact location was still high in
the leafless season although the risk was reduced compared to general locations. The ratio of throughfall
precipitation at the concentrated impact location and at general locations compared to freefall
precipitation were 8.1 and 0.71, respectively, suggesting that throughfall precipitation widely decreased
with canopy interception whereas the identified hotspot of throughfall selectively increased it. Each slope
of the relationships between TKE and throughfall precipitation at the concentrated impact location and
general locations was higher in the leafed season than in the leafless season (ANCOVA, $P < 0.01$).



Therefore, we can conclude that unit TKE strongly increases with the presence of leaves and potential
splash erosion is higher during the leafed period. However, unit TKE at the concentrated impact location
in the leafless season (36.4 J m$^{-2}$ mm$^{-1}$) was still higher than at general locations in the leafed season (32.1
± 10.3 J m$^{-2}$ mm$^{-1}$). This suggests high splash erosion risk at the concentrated impact location even in the
leafless season. In summary, leaf status has shown to generate a distinct impact and differentiation of
effects, and the second hypothesis can therefore be accepted.

193        Additionally, differences between TKE and FKE as well as throughfall and freefall precipitation

appear to be less pronounced in the leafless season. Levia et al., (2019) showed canopy drips under
broadleaved trees accounted for 69% of total throughfall precipitation in the leafed phenophase,
compared to 8% in the leafless phenophase. Most of the throughfall at general locations under leafless
trees were composed of freefall. Soil erosion risk is less during leafless season than leafed season except
for the concentrated drop impact locations.

**3.3 Implication and uncertainty**
This study remarked notably high TKE under investigated beech trees. Mean unit FKE has been reported
by van Dijk et al., (2002) calling 14.2, 18.6, 26.5, and 28.1 J m$^{-2}$ mm$^{-1}$ with rainfall rates of 1, 10, 50, 100
mm h$^{-1}$, respectively. The measured maximum unit FKE was 28.3 J m$^{-2}$ mm$^{-1}$. As for throughfall, unit
TKE reported in previous studies ranged from 16.4 to 28.1 J m$^{-2}$ mm$^{-1}$ in Japan (Nanko, 2013), Hawaii
(Nanko et al., 2015) and Thailand (Nanko et al., 2020). The unit TKE at the concentrated impact location
in the present study was much higher than these previously reported values. The high TKE induced by not
only throughfall precipitation, but also larger throughfall drop size distributions, resulted in an increased
risk of soil erosion. Furthermore, unit TKE for general locations in the present study was also higher than
in previously measured Japanese cypress plantations with 16.4 - 21.0 J m$^{-2}$ mm$^{-1}$ (Nanko, 2013). The
median volume drop size of canopy drip from leaves was 4.7 mm in Japanese cypress but 5.2 mm in
beech (Nanko et al., 2013). This difference was caused by varying leaf traits such as leaf area, leaf shape,
and leaf surface water repellency (Levia et al., 2017). Thus, TKE generation is strongly species specific
and TKE under beech trees may be higher than under other tree species.

214        Finally, although considerable higher TKE at the concentrated impact location was measured

using splash cup, we should note that TKE at the concentrated impact location in the present study may
be underestimated due to the rim effect related to the splash cup measuring system. There is some
uncertainty in the estimated TKE if sand particles are starting to hit the cup wall instead of flying out.
This phenomenon occurred especially at the concentrated impact location. Thus, TKE at the concentrated
impact location may be even higher than reported TKE in the present study.

**4. Conclusions**
In this paper, we report results from a splash cup experiment to investigate potential erosion from high
energy water release points under the canopy in a disturbed Japanese forest environment. Extremely high
TKE was observed from structurally-mediated woody surface points under beech (*Fagus crenata*)



showing values approximately 15 times higher than at general drip locations and approximately 50 times
higher than FKE. The higher kinetic energy was caused by both higher throughfall precipitation and
higher unit kinetic energy. These results underline the evidence of high soil erosion risk in forested areas
due to particular tree traits and show that this risk can significantly exceed the previously known
dimensions at specific points under the tree canopy. Moreover, unit TKE at high-energy and general
locations was reduced in the leafless season, but unit TKE in the leafless season was still higher at the
concentrated impact location than at general locations in the leafed season. This result points to a
potentially enhanced soil erosion risk even outside the growing season if concentrated impact locations
with high kinetic energies occur in larger numbers on trees. Further research is necessary to verify the
results, expand them to include other tree species and forest ecosystems and to shed more light into
mechanistic effects of distinct plant characteristics. In this context, it should also be investigated how
many of these concentrated impact locations may occur on average on different tree species to better
assess the extent of the erosion risk. This becomes particularly important when the protective soil cover
layer with understory or leaf litter is disturbed or removed. Therefore, future studies examining soil
erosion rates under forests need to considerate both changes in TKE through plant traits and variations in
ground cover.

**Data availability**
All raw data is provided in the supplement material.
**Video supplement**
https://doi.org/10.5446/61199
**Author contribution**
AK, KN and SS designed the experiment, AK, YS, TK and SJ carried it out. AK, KN and SS prepared the
manuscript with contributions from all co-authors.
**Competing interests**
The authors declare that they have no conflict of interest.
**Acknowledgments**
We thank the technical staff of Shiiba Research Forest who helped with the preparation and establishment
of measurements. We also thank Kyushu University Fund which allowed us to meet in Shiiba.
**Financial support**
This study was financially supported by JSPS KAKENHI Grant Number 22H03793 and JSPS
Postdoctoral Fellowship for Research in Japan (Short-term) Grant Number PE21018.

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




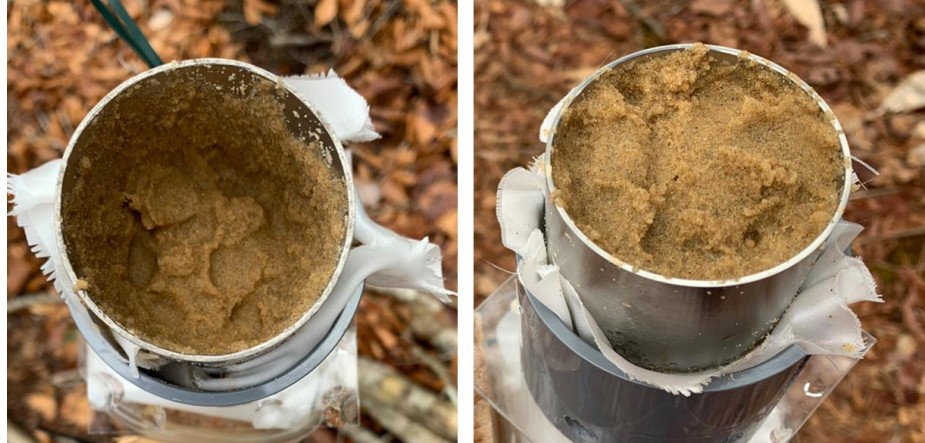



**Fig. 1** Splash cups at the concentrated impact location (left) and at an exemplary general location (right)
after the first rainfall event in the leafless season. Freefall precipitation of this event was 35.4 mm.

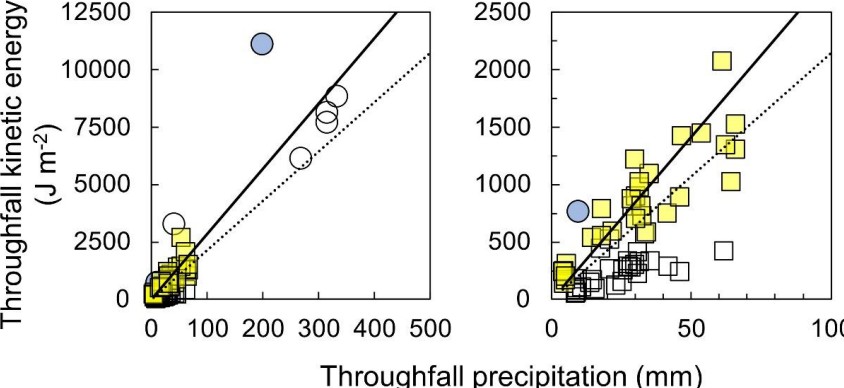



**Fig. 2** Relationship between event-based throughfall precipitation and event-based throughfall kinetic
energy (TKE). Circles and squares show TKE measured at each concentrated impact location and each
TKE at general locations, respectively. Closed and open symbols show leafless and leafed seasons. Solid
and dotted lines show the regression lines at the concentrated impact location and general locations,
respectively. The relationships were significantly different between the locations (ANCOVA, $P < 0.01$).






**Table 1** Event-scale precipitation, kinetic energy, and unit kinetic energy at the impact location and
general locations under Japanese beech trees and outside the forest in the leafless and leafed seasons,
respectively.

| Duration | Precipitation (mm) | | | Kinetic energy (J m$^{-2}$) | | | Unit kinetic energy (J m$^{-2}$ mm$^{-1}$) | | |
|---|---|---|---|---|---|---|---|---|---|
| | Impact locations | General locations | Freefall | Impact locations | General locations | Freefall | Impact locations | General locations | Freefall |
| Leafless | | | | | | | | | |
| 3/3-7 | 331.7 | 26.1 ± 8.9 | 36.0 ± 0.4 | 8869 | 274 ± 157 | 161 ± 20 | 26.7 | 11.5 ± 8.5 | 4.5 ± 0.5 |
| 3/11-13 | 40.4 | 9.1 ± 0.8 | 11.9 ± 0.2 | 3307 | 102 ± 43 | 48 ± 2.9 | 81.9 | 11.2 ± 4.7 | 4.0 ± 0.3 |
| 3/19-22 | 314.4 | 37.1 ± 14.0 | 43.4 ± 0.7 | 7737 | 396 ± 166 | 385 ± 77 | 24.6 | 9.5 ± 2.3 | 8.9 ± 1.9 |
| 3/27-29 | 314.4 | 31.0 ± 7.3 | 38.8 ± 0.7 | 8166 | 387 ± 222 | 294 ± 19 | 26.0 | 13.1 ± 8.1 | 7.6 ± 0.4 |
| 4/3-5 | 268.2 | 20.5 ± 8.5 | 24.8 ± 0.2 | 6182 | 291 ± 188 | 25 ± 11 | 23.1 | 13.8 ± 6.9 | 1.0 ± 0.5 |
| Leafed | | | | | | | | | |
| 8/19-21 | 445.3[a] | 39.1 ± 12.9 | 54.1 ± 1.3 | 11571[a] | 893 ± 189 | 561 ± 47 | 26.0 | 24.2 ± 7.6 | 10.4 ± 0.9 |
| 9/2-3 | 9.4 | 4.5 ± 0.5 | 5.1 ± 0.3 | 769 | 223 ± 63 | 27 ± 8 | 81.6 | 49.7 ± 13.7 | 5.2 ± 1.3 |
| 9/10-16 | 797.5[a] | 56.9 ± 7.3 | 97.0 ± 1.4 | 20723[a] | 1723 ± 560 | 322 ± 50 | 26.0 | 30.9 ± 11.4 | 3.3 ± 0.5 |
| 9/27-10/1 | 498.6[a] | 38.8 ± 14.6 | 60.6 ± 1.9 | 12955[a] | 1014 ± 303 | 7 ± 1.4 | 26.0 | 27.4 ± 7.9 | 0.1 ± 0.0 |
| 10/8-11 | 223.7[a] | 22.0 ± 7.9 | 27.2 ± 1.5 | 11137 | 706 ± 186 | 12 ± 5.7 | 49.8 | 33.3 ± 7.7 | 0.5 ± 0.2 |

Data are given as mean ± standard deviation.
[a] The data was estimated from freefall precipitation.