# Peer review of "Short communication: Concentrated impacts by tree canopy drips: hotspots of soil erosion in forests"

_Earth Surface Dynamics, 2023_

## Author Comment (AC1)

*Reviewer #1*

*This manuscript reports some estimates of throughfall kinetic energy at the soil surface. The motivation for the report is that very high kinetic energies were measured at a selected location where there was concentrated drip from a tree. The experiment tests a trivial hypothesis L85 L162. We already know that throughfall energy is greater when and where there is more throughfall and when and where drops are larger. The effect of leaf status is more interesting.*

*I do not think this sampling effort is sufficient for the data to be reliable. L113 one drip point and 6 non-drip points, measured in 10 rainfall events is a very small sample size, given that spatial variability of throughfall is usually very high. Also, measurements at the single concentrated throughfall point failed in all but three events. It appears (L144) that there was even some pseudoreplication in the ANCOVAs because regression was used to obtain throughfall estimates in four events (L122). Finally, the gap-filled data were all in the leafed period, and the conclusion is that the unit kinetic energy during the gap-filled events was about half or less than the one measured event (Table 1)—this is an example of how poorly constrained the estimates are.*

*All estimates of the magnitude of energy concentrated at drip points depend on overinterpretation of data. With one, non-randomly-placed, drip-point sampler, we do not know how widespread these points are nor how variable they are, so there is no way to make any estimate of the importance of drip points at any scale.*

We want to thank reviewer # 1 for these critical comments. We agree with the reviewer's opinion on high spatial variation in TKE and that the dataset presented here in this short communication would not be sufficient to describe the entire splash phenomenon in our study area. To be precise, we measured TKE at 36 locations to evaluate spatial variation in TKE in the same forest and this dataset will be used to describe variability of general splash locations according to tree traits.

However, our objective in the present study is to evaluate the possible maximum level of TKE under the forest, which is defined by concentrated drip points. This phenomenon has been ignored so far and high TKE values were mostly excluded from datasets as outliers in all studies we are aware of. Therefore, it has never been reported in previous studies and no specific numbers of the maximum kinetic energy have been published (please indicate if otherwise). From a soil erosion researchers' perspective, we believe it is of great interest to obtain a better description of the phenomenon before measuring more precisely with higher technical and financial effort. Therefore, and after discussing with other researchers from the same field, we decided to report the intensity of a concentrated impact location compared to general locations to bring the focus on this drip point phenomenon under tree

canopies as a short communication. We apologize if our intention did not become clear from the manuscript draft.

As the reviewer suggested, it is very important to know how widespread and variable these impact locations are. Also, we agree that leaf status is very important and even less investigated. We have to answer this question as a next step and will conduct further research on that. In this context, we must point out that finding suitable measurement locations and precisely place splash cups under concentrated drip in the forest is anything but trivial. Therefore, a fully randomized sampling design, as we typically use it in our splash studies so far, might not be suitable. We have been somehow fortunate to find and reliably maintain a suitable measurement point over a longer period in the presented study.

Finally, we agree that this statistical analysis is less powerful with limited data, but we do not consider it inaccurate. Precipitation and TKE at the impact and general locations were used for the ANCOVA. We estimated precipitation at the impact location at one sampling period using freefall precipitation outside the forest which was not included in the ANCOVA. Thus, we do not believe that the analysis is pseudoreplicated.

*Minor comments*

*Fig 2 I think the right-hand panel is a blowup of the left but there are no labels to support this guess. It would be much easier to read this figure if there were labels instead of text to describe the symbols.*

Response

As the reviewer suggested, the left-hand graph was zoomed from the right-hand graph. We will improve the Figure by inserting lines between the panels and adding a legend with labels.

*L153-154 but the branch height for the concentrated drip point was the same as the others and leaves were not measured by location, so the experiment did not address these questions.*

Response

As the reviewer suggested, the description was inaccurate. The sentence will be removed.

*L164-171 the points about terminal velocity do not lead to the conclusion L169.*

Response

We are sorry for the poor explanation in the previous manuscript. We will add the explanation that raindrop could not reach terminal velocity and branch height was one of the factors determining TKE in the present study, but the higher TKE at the impact location was not induced by the branch height

because of the comparable branch height at the impact location with other general locations.

*L181 what is risk exactly, and how can it be lower at the drip point than elsewhere?*

Response

We are sorry for the inaccurate description. We will replace "general locations" with leafed season.

*Table 1 column headers say "Impact locations" but there was only one.*

Response

We are sorry for the mistake. We will remove "s".

*L207 there are no drop-size distribution data presented. I think the inference is correct but the wording must careful not to imply this research supports the statement directly.*

Response

We really appreciate the valuable comment. We will revise it carefully.

*Th English could be improved substantially. Some problems: L46 punctual is not the right word; L50 grazing (although grazing is probably too specific of a word and "feeding" would probably be better); L71 differs; L73 "is different"; L83 occurred with splash cups? L103 sentence makes no sense; L104 weighed dry; L118 "failed" instead of "missed"; "was," not "were"; L140 ANCOVA does not only examine significant differences; L149 considerably; L159 "The It"; L167 do not reach terminal; L178 than that at; L183 widely?*

Response

Thank you for the comment. The current manuscript will be edited by a professional English editor.

---

## Author Comment (AC2)

*Reviewer #2*

*This study measured the throughfall kinetic energy (TKE) and throughfall precipitation at concentrated and general locations beneath canopies of the Shiiba research forest in Japan, which were compared with freefall kinetic energy (FKE) and freefall precipitation at both leafed and leafless stages, respectively. Thus, the splash erosion caused by droplet impacts could be investigated, accordingly. The authors found that TKE at the concentrated impact location was 15.2 and 49.7 times higher than that at general locations under beech canopies and FKE, respectively. This study confirmed that canopy drip from woody surfaces can be a hotspot of soil erosion in temperate forest ecosystems. The potentially high rates of sediment detachment could be induced by not only throughfall precipitation but also larger throughfall drop size distributions at concentrated impact locations. This topic is of scientific significance, and falls in the research scope of Earth Surface Dynamics. I recommend accepting this study after the below-mentioned revisions have been addressed.*

Response

We really appreciate the reviewer's positive evaluation. We read the comments carefully and revise it according to the comments.

1. *Recommend to add a figure in Section 2 to show the concentrated and general locations for measuring throughfall, and the location where the freefall was measured. It benefits a clear introduction of experimental design in this study.*

Response

Thank you for the valuable suggestion. We will add the figure showing the location of splash cups and rainfall collectors.

2. *Detailed descriptions of the splash cups, such as their diameter, height, etc., are needed, because these cup characteristics affect the quantitative measurements of loss of soil (LOS) and consequent TKE via linear regression.*

Response

We will add the information; diameter and height of the splash cup were 5.0 cm and 5.1 cm, the volume was 100cc. These are slightly larger than those reported by Scholten et al., 2011 (4.6 of diameter and 3.6 cm of height, respectively), but accurately estimated TKE by using a linear equation (Shinohara et al. 2018).

3. *Lines 121–124: There were no introductions on how to get these quantitative relations of freefall precipitation with TKE and throughfall precipitation. If doing regressions based on the*

*measurements in this study, please add the data and analysis. If citing other research, add the references, please.*

Response

We will add the information. The data was obtained in this study. We obtained data of 10 events (Table 1), but TKE and throughfall precipitation at the impact location were obtained in seven and six events. Thus, the relationship between TKE and freefall precipitation was established using the data obtained in seven events whereas the relationship between throughfall precipitation and freefall precipitation was established using the data obtained the six events.

4. *The authors installed seven splash cups to measure TKE, with six cups at general locations and one cup at possible concentrated location. However, throughfall measurements were not clearly described in this study. Is it that throughfall precipitation and TKE were measured at the same location? If so, how to precisely measure TKE by using the splash cup and avoid the disturbance of throughfall precipitation measurements at the same time and locations via installing rain gauges?*

Response

We will add the explanation about the throughfall measurement. A storage-type bottle with a funnel (diameter: 9.0 cm) was installed next to each splash cups to measure precipitation. Precipitation was measured at the same time with TKE measurement. The distance between the splash cup and precipitation collector was about 20 cm, thus, the location of throughfall precipitation was not exactly same with the splash cup.

5. *The authors measured tree traits, such as diameter at breast height, tree height, LAI, leaf area, leaf mass per area, etc. They particularly addressed the effects of structurally designed high energy points on TKE in Section 3.1. However, there were no quantitative descriptions to introduce what is the structurally designed high energy points like, and no quantitative analysis to defend the claim of its effects on TKE.*

Response

We agree with the reviewer's suggestion that it is interesting to examine the effect of tree traits on TKE under the canopies. However, we did not examine it in the present study because we just showed the tree traits to characterize the studied beech tree, not to examine spatial variation in TKE under the beech canopy. We measured TKE and throughfall precipitation under 10 different tree species in this forest to examine the effect of tree traits on TKE and prepare a manuscript as another paper. Thus, we focus on the TKE at the impact location and does not examine spatial variation in TKE with tree traits.

6. *The authors discussed the effects of leaf status (i.e., leafed and leafless) on TKE and consequent splash erosion risks. They conducted these measurements in spring and summer from March 3rd to April 5th, and in autumn and winter from August 19th to October 11th, respectively. However, in addition to the influence of different leaf statuses, the distinct meteorological conditions also significantly affected throughfall precipitation and TKE. Therefore, the authors might need more evidence to support their claim that leaf status, not the meteorological conditions, dominated the influence on splash erosion risks.*

Response

We agree with the reviewer's suggestion that meteorological condition, such as intensity and amount of precipitation, can affect TKE. Although we did not monitor temporal changes in throughfall precipitation within rain events at the study location, we have been measuring 10-minutes open space precipitation at the University Forest office, situated 4km away from the study site [600 m a.s.l.]. There was high variation in precipitation amount among the months and considerable high precipitation was observed in August. In Japan, it is higher precipitation in the summertime because of rainy and typhoon season. Precipitation amount is the most important factor determining soil erosion risk and precipitation amount in the leafless season after soil thawing is relatively less than that in the leafed season in Japan. We will add some discussion relating the effect of meteorological condition. We will also add such kind of meteorological data in the site description section.

**Table** Precipitation data at the University Forest Office in 2021.

| | Precipitation amount (mm) | Number of precipitation event | Precipitation intensity during rainfall event (mm $h^{-1}$) | Number of erosive precipitation events (>12.7mm event$^{-1}$) |
|---|---|---|---|---|
| March | 162 | 9 | 1.44 | 4 |
| April | 133.5 | 8 | 1.30 | 3 |
| August | 958.5 | 15 | 2.67 | 7 |
| September | 170 | 11 | 1.40 | 4 |
| October | 41.5 | 6 | 1.47 | 0 |

*Minor suggestions:*

*Line 85: No need to start a new paragraph to state the hypotheses.*

*Line 159: Delete "The" before "It".*

Response

Thank you for the suggestion. We will revise them.

---

## Author Response (AR2)

Dear Dr. Sagy Cohen,

We really appreciate your understanding and constructive suggestion. We have added an explicit explanation of the limitation in the abstract, results and discussion and conclusions sections according to your comment described below. We believe that the current manuscript is now ready for publication. We are looking forward to hearing from you.
* * *
*Editor's comment*
*We received two excellent reviews with somewhat conflicting recommendations. The authors provided a strong response and addressed most of the reviewers' comments. I applaud the authors for addressing the reviewers' more technical comments. Considering that this paper is a 'Short Communication', I have found the authors' response to the critique (particularly by reviewer #1) about the research design and data collection extent to be reasonable. However, they must be clearly articulated before the manuscript can proceed to publication. In the revised manuscript, please add an explicit expression of the shortcomings and limitations of the data and collection strategies as raised by the reviewers. Since these shortcomings need to be clear to the reader, I would like to see these added to the abstract, results & discussion, and conclusions sections.*
* * *
Yours sincerely,
Ayumi KATAYAMA